# Empathy as a Factor Conditioning Attitudes towards the Elderly among Physiotherapists—Results from Poland

**DOI:** 10.3390/ijerph19073994

**Published:** 2022-03-28

**Authors:** Marta Podhorecka, Anna Pyszora, Agnieszka Woźniewicz, Jakub Husejko, Kornelia Kędziora-Kornatowska

**Affiliations:** 1Department of Geriatrics, Faculty of Health Sciences, Collegium Medicum in Bydgoszcz Nicolaus Copernicus, University Copernicus, 85-094 Torun, Poland; awozniewicz@cm.umk.pl (A.W.); kubahusejko@gmail.com (J.H.); kornelia.kornatowska@cm.umk.pl (K.K.-K.); 2Department of Palliative Care, Faculty of Health Sciences, Collegium Medicum in Bydgoszcz Nicolaus Copernicus, University Copernicus, 85-094 Torun, Poland; anna.pyszora@cm.umk.pl

**Keywords:** ageism, elderly, empathy, physiotherapists

## Abstract

Attitudes of healthcare workers towards the elderly significantly affect the quality of care dedicated to this group of patients. The purpose of this study was to assess the approach of Polish physiotherapists to the elderly and to analyze the factors influencing these attitudes. The study was based on a cross-sectional Internet-based survey that was conducted in the period from May to September 2021. It was completed by 252 subjects: 189 women and 63 men. The study was based on the involvement of physiotherapists with the right to practice their profession in accordance with the law in force in Poland. The tools used in the study were The Kogan Attitudes towards the Elderly (KAOP) score and The Jefferson Empathy Scale (JSE). To model the KOAP score as a function of the predictors, Bayesian linear regression was used. The average KOAP score in the sample was M (SD) = 100.7 (17.46), with the central 50% of the observations ranging from 81 to 113 points. We observed one statistically credible relationship: relevance of contacts with elderly people was positively and moderately related to KOAP. Additionally, we observed that the JSE scale was positively, but very weakly, associated with KOAP. Empathy and own experience of health care providers may protect against negative attitudes towards the elderly, meaning empathy in healthcare professionals is crucial.

## 1. Introduction

The phenomenon of exclusion on the basis of advanced age was first mentioned in 1969 by Robert Butler who used the term “age-ism” in an interview for the Washington Post, describing the negative attitude of residents of Chevy Chase in Maryland, USA, towards the policy of facilitating living for seniors in their town [1]. He then described his observations in *The Gerontologist* journal, in which he predicted that the problem of the exclusion of the elderly would, in 20–30 years, become what racism was in the USA in the 1960s [2]. These words are now being confirmed, when, despite the efforts of many researchers to find a solution leading to the reduction of the phenomenon of exclusion of the elderly, this problem is still noticeable, and social awareness of the existence of ageism is lower than in the case of, for example, sexism or racism [3].

Further research on the phenomenon of ageism is necessary due to the aging of the societies of developed countries. Demographic forecasts regarding the share of people of advanced age in the populations of the above-mentioned countries indicate that the problem related to this phenomenon will become more and more noticeable. The scale of the discussed process is evidenced by the creation of the concept of “aging Europe”, summarizing the prevailing decline in fertility, reduced mortality, and longer life expectancy in European populations [4].

In the situation of an aging society, the environment related to the healthcare system, and, in the context of ageism, the attitude of the staff cooperating with seniors, plays a special role. Unfortunately, numerous studies measuring the phenomenon of negative attitudes towards the elderly in healthcare facilities have shown the existence of under- or even over-treatment for the elderly, because too many clinicians confuse acute diseases with the physiological aging process of the body. It is also pointed out that medics ignore the pain, anxiety, or depression reported by patients and interpret them as inevitable features of advanced age. One can also often find, even subconsciously, the perception of seniors as being less worthy of attention than younger patients, who are faced with the prospect of more years in the future if cured [5].

The perception of seniors by medics cannot be considered to be a phenomenon that does not change over time, especially in times when efforts are made to change the habits of the elderly. The 2002 report “Active Aging: A Policy Framework” published by the World Health Organization, promoted the importance of activity in the aging process in a multifaceted approach, taking into account the necessity of numerous changes; for example in the health and social service system, personal or economic factors [6]. In 2010, in turn, attention was drawn to the need to inform the public about the importance of physical activity in the document “Toronto Charter for Physical Activity: A Global Call for Action” [7]. The promotion of this trend in the media was not as noticeable as expected [8], but people associated with healthcare systems can better understand the assumptions of these documents.

Under the conditions of systemic promotion of active aging, there may be discrimination against overweight and obese elderly people who, through stereotyping, are associated with non-compliance with the above-mentioned trend. Research conducted on medical students in the USA showed that students perceive obese people as ugly, lazy, sloppy, and more depressed [9]. Similar results were obtained in studies of students from Germany [10] and Turkey [11], and studies on both nursing students and registered nurses from China showed that professionally active people had more negative attitudes towards obese people than those still in education [12]. However, no analogous studies on discrimination with regard to the active aging trend among physiotherapists have been conducted.

A much more current problem that may have significantly influenced the perception of seniors by healthcare professionals is the COVID-19 pandemic. It is important to highlight that elderly people are most exposed to severe courses of infection with SARS-CoV-2 virus [13]. This relationship was previously demonstrated in reports from January 2020 [14], which allowed for the rapid development of a strategy aimed at protecting seniors against the discussed coronavirus. However, appropriate strategies have repeatedly and publicly noted the special importance of protecting people in advanced age [15], which may have influenced the perception of all seniors by medical professionals as requiring special care. This type of reasoning, despite the willingness to help, is associated with stereotyping and may lead to the development of the phenomenon of ageism.

Research on the attitude of healthcare workers, such as physiotherapists, to the elderly is particularly important due to the importance of anti-ageist attitudes in the quality of treatment of seniors. In the studies carried out by Wolff et al., in which attempts were made to determine the positive views that patients have of themselves in terms of the effectiveness of their physical activities, it was shown that the positive attitude of the elderly respondents correlated positively with the effectiveness of the introduced intervention [16]. Due to the small number of reports in this regard, it is important to determine whether the socio-demographic parameters have an impact on the attitudes towards the elderly among physiotherapists. Additionally, when designing such a study, one should rely on research carried out on other professional groups related to health protection, where intergenerational contacts were also an important issue [17]. The results of these studies are of particular importance in relation to physiotherapists, because it is part of their work to introduce appropriate physical exercises. For this reason, research into the possible presence of the phenomenon of ageism among physiotherapists has practical justification.

In order to study the attitude of a specific professional group in relation to people in advanced age, it is necessary to identify factors that may have a potential impact on the development of ageism. Gender may be such a factor, as it has already been shown that the prevalence of ageism in men is statistically higher than in women [18]. Age can also be taken into account as a possible factor, as it has already been shown that it may have an impact on ageism, in addition to the length of service in an occupation based on working with an elderly person [19]. Studies conducted in Poland have shown that living in a large city or in the countryside affects ageism [20]; therefore, this factor should also be taken into account. It is also important to take into account the level of education in the research, because, as demonstrated by the example of nurses in Jordan, the low level of education in the field of dealing with seniors significantly increases the level of ageism [21]. Ultimately, marital status can be considered as a potential factor that may influence ageism. Although this has not been demonstrated, it has been found that single people are more likely to have ageistic thoughts [22], and, therefore, the development of this correlation is justified.

The aim of the study was to find scientifically based answers to the following questions:What is the attitude of physiotherapists towards the elderly?Can the attitudes towards the elderly among physiotherapists depend on: age, gender, place of residence, level of education, marital status, or duration of employment?Can physiotherapists’ attitudes towards the elderly depend on regular contact with the elderly?Can attitudes towards the elderly depend on the empathy shown by physiotherapists?

## 2. Materials and Methods

### 2.1. Purpose

A cross-sectional Internet-based survey was conducted in the period from May to September 2021. Participation was anonymous and voluntary. The study was based on the involvement of physiotherapists with the right to practice their profession in accordance with the law in force in Poland [23]. The online questionnaire was developed by a research team via an Internet platform (Survgo system). Only adults were allowed to participate in the study, with no upper age limit. All participants were informed that their responses were anonymized. Only full responses were intended for analysis.

The study was approved by the Bioethics Committee of the Nicolaus Copernicus University Collegium Medicum in Bydgoszcz, Poland (KB 83/2021), and was conducted in accordance with the Helsinki Declaration.

### 2.2. Participants

The survey was completed by 252 subjects: 63 men and 189 women. The total number of physiotherapists in Poland is estimated to be 71,471, comprising a majority of women (70%) [24].

### 2.3. Methods

The first form contained a socio-demographic questionnaire on marital status, place of residence, level of education, and place and length of service. In addition, questions were asked about contact with patients/people over 65 years of age, during work and privately.

The Kogan Attitudes Elderly scale (KAOP) was used to assess the attitudes towards the elderly [25], containing 34 questions with six-point answers (1 = strongly disagree; 6 = strongly agree). The possible scores ranged from 34 to 204, where higher scores indicate a more positive attitude towards the elderly, and lower scores mean a more negative attitude towards older people [26]. The KAOP scale had been previously translated and adapted to Polish conditions, and studies with its use in Poland have also been conducted [26], but it has not been validated and compared with other language versions.

The Jefferson Empathy Scale (JSE) was used to self-assess empathy by the respondents [27]. It was developed by Hojat et al., and has already been used in over 80 countries globally. The scale consists of 20-point questions, with 7-point answers (1 = strongly disagree, 7 = strongly agree). Half the items are positively worded and scored directly, and the other half are negatively worded and scored in reverse order. For the study of physiotherapists, we used the JSE-Physician/Health Professions (HP-version) [28]. The Polish version of JSE, used in our study, was validated, and its Cronbach’s α reliability coefficient was 0.79 for the HP-version [29].

Due to the ongoing pandemic, the research was carried out via the Internet. The questionnaires were disseminated via social media and via e-mails from the National Chamber of Physiotherapists. The forms were placed on the Survgo platform.

### 2.4. Data Analysis

We used R 4.0.2 (Bell Laboratories, Murray Hill, NJ, USA) to analyze the data [30]. To model the KOAP score as a function of the predictors, the Bayesian linear regression was used. Ordered categorical predictors were coded with orthogonal linear contrast, and unordered categorical predictors were coded with sum-to-zero contrast. Continuous predictors were entered into a model on a standardized scale.

The inference was based on analyzing the posterior probability distributions of a model parameters, obtained by integrating likelihood with prior probability distributions. Regression weight is statistically credible when 95% credible intervals (95% CI) of the posterior distribution exclude zero [31]. The mean of the posterior distribution is presented as a point estimate of the effect. Default improper flat priors were used for the regression weights.

Predicted marginal means with corresponding 95% CI are presented in the figures to investigate the relationship between dependent variable and a credible predictor. These values represent a median of a posterior distribution of KOAP values.

The Markov Chain Monte Carlo (MCMC) sampling procedure was conducted, using the brms package, to approximate the posterior distributions of the models [32]. Six parallel chains were used, each consisting of 6000 samples (with 3000 samples used as warm-up period and every 10th sample recorded, resulting in 1800 recorded samples in total). The resulting sample was well-mixed and did not contain autocorrelated chains or unimodal posteriors. Model accuracy was assessed with posterior predictive checks.

## 3. Results

The characteristics of the participants (*n* = 252) are listed in Table 1.

The average KOAP score in the sample was M (SD) = 100.7 (17.46), with the central 50% of the observations ranging from 81 to 113 points. The KOAP score reliability was high, and Cronbach’s α = 0.86. The average JSE score in the sample was M (SD) = 106.31 (12.38). Model coefficients of the linear regression with KOAP score as the dependent variable are summarized in Table 2, whereas model predictions for the credible predictors are presented in Figure 1. We observed one statistically credible relationship: relevance of contacts with elderly people was positively and moderately related to KOAP score. Additionally, we observed that JSE scale was positively, but very weakly, associated with KOAP score (note that the lower credible interval is zero). The remaining relationships were not statistically significant.

## 4. Discussion

Empathy and own experience of healthcare providers may protect against negative attitudes towards the elderly. According to the fact that seniors are most exposed to severe courses of infection with the SARS-CoV-2 virus [13], we considered it essential to analyze ageism among healthcare providers during the pandemic. Healthcare providers’ knowledge and attitudes about aging can affect how accurately and sensitively they distinguish between normal aging changes and acute or chronic disease changes [5]. An important component in research on ageism is the level of empathy [33,34,35]. Empathy is an important part of high-quality healthcare for the elderly [25,26] and is negatively associated with ageism [35].

The aims of the study were achieved. In our research, physiotherapists’ attitudes toward older adults proved to be positive. Relevance of contacts with elderly people and level of empathy of physiotherapists were positively related to attitudes towards the elderly. The results showed no differences between attitudes toward older adults and age, gender, place of residence, level of education, marital status, or duration of employment among surveyed physiotherapists. It should be noted, however, that in terms of age, the attitude towards people in advanced age can also be considered in the context of fear of their own aging and death. A study on these issues was carried out by Kolushev et al., where it was shown that ageism increased with increasing fear of aging among respondents, and decreased with increasing fear of death [36]. Our study did not take these issues into account.

The lack of influence of the place of residence on ageism is also not justified in previous studies. According to the measurements carried out in Poland, every fifth elderly person living in the countryside and as many as every third person living in the city face the problem of discrimination [20], which is a significant difference.

Earlier publications indicated that ageism was more common among people with lower education and those having a poorer level of knowledge about human aging [37]. This information is important due to the lack of correlation between the level of education and ageism in our study.

In studies conducted among doctors and nurses from Russia, a weak but statistically significant correlation was found between ageism and duration of employment [19]. The lack of dependence between the aforementioned parameters among physiotherapists obtained by us must therefore be confirmed in subsequent studies in order to be able to draw more certain conclusions.

In the available literature, no previous correlation between ageism and marital status among physiotherapists was found.

In the population of professionally active physiotherapists, there are no articles about attitudes towards the elderly, and no analysis of the factors influencing these attitudes has been performed. One study we found, from the 17th International World Physical Therapy Congress, found that attitudes of physiotherapists working in Singapore, in acute inpatient settings toward older adults, were generally positive [38]. Two characteristics were found to be associated with the attitudes toward older adults: the ability to communicate effectively and spending more time with older adults (measured by Modified University of California at Los Angeles Geriatrics Attitudes Scale) [39]. However, such studies were conducted among physiotherapy and medical students.

In a systematic review of 14 studies, Kalu et al. emphasize that students of physiotherapy have a positive attitude towards the elderly. Four of the analyzed studies showed that physiotherapy students were interested in working with seniors, three of whom indicated low interest and one was moderate. It is worth emphasizing that contact with older adults before starting physiotherapy education was found to be the only factor that showed a positive impact on attitudes towards the elderly [40]. Similar observations were made by Acikgoz et al. Their findings suggest that interactions with older adults may increase age-related positive attitudes among physiotherapy students [41]. To assess ageism attitudes, authors have also used The Ageism Attitude Scale [42].

Educational strategies can influence the empathy of physiotherapy students, so it is worth using these strategies during the learning process for future health providers [43,44]. In a study in which first-year medical students participated [43], the instruments used were UCLA Geriatric Attitudes Test [39] and Modified Maxwell–Sullivan attitudes towards the elderly scale [45]. Among undergraduate nursing students [44], KAOP, the same tool used in our study, was used to assess the attitudes towards the elderly. Healthcare professionals should be supported through lifelong learning and personal development programs, in addition to supervision sessions that will enable them to develop empathic skills [46]. In research among medical students, the most commonly used instruments have been the Aging Semantic Differential and the University of California Los Angeles Geriatric Attitudes Scale; KOAP has also been used. Medical students showed neutral or positive effects of geriatric medical curriculum innovation on their attitudes [47].

In our study, as many as 75% of all respondents were women, which may have significantly influenced the results obtained. In the study conducted by Bodner et al., which assessed the tendency to discriminate against the elderly, depending on the sex and age of the respondents, a more frequent occurrence of ageism was noticed in men than in women [18]. Taking into account the fact that, in our study, the attitude towards people in advanced age was found to be generally positive, further studies in the future, with a higher percentage of men as respondents, may be necessary for comparative purposes.

The healthcare environment is not immune to the harmful effects of ageism. Ageism unfortunately still exists among the attitudes of health providers, and can be explicit or implicit. Numerous interventions are underway that should begin to alleviate ageism. The feeling of empathy among health care professionals is crucial.

However, this research has several limitations. The main limitation of the study is the unequal gender distribution in the study group. It is worth emphasizing, however, that the study distribution corresponds to the distribution in the population of Polish physiotherapists. In addition, in future research, it would be worth assessing the factors that affect the attitudes towards the elderly presented by physiotherapists. The project was carried out online, which may also have affected the results obtained. In addition, we conducted research during a pandemic, which may have temporarily affected the views of medics. In the future, we would like to repeat the project outside the pandemic period to finally confirm the results obtained.

## 5. Conclusions

In our research, the relevance of contact with elderly people and the level of empathy demonstrated by physiotherapists were positively related to attitudes towards the elderly.

We hope that our study will inspire other researchers. There is a need for further observations to help determine the factors that influence the formation of specific attitudes towards the elderly. Identifying the determinants of specific attitudes may contribute to building more effective educational programs for students and already working health care professionals. Knowledge of empathy, in addition to training in demonstrating it, can be effective in reducing ageism among service providers. Additionally, an interesting research direction may be an attempt to determine the relationship between ageism and empathy among citizens of different culturally diverse countries. This is especially important given the growing number of elderly people in most societies.

## Figures and Tables

**Figure 1 ijerph-19-03994-f001:**
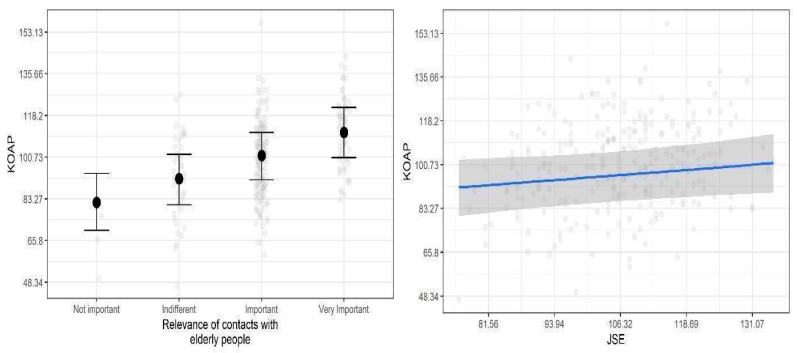
Posterior medians (points and blue line) of the predicted mean KOAP scores as a function of credible predictors. Vertical lines and shaded areas are 95% credible intervals. Grey transparent points show data. Note: relationship between KOAP and JSE.

**Table 1 ijerph-19-03994-t001:** Characteristics of participants (*n* = 252).

Variable	Gender	Frequency	%
Gender	female	189	75
	male	63	25
Age	18–29 years	80	31.75
	30–39 years	105	41.67
	40–49 years	47	18.65
	50 years and more	20	7.94
Place of residence	village	51	20.24
	city up to 50 k inhabitants	38	15.08
	city up to 100 k inhabitants	33	13.1
	city up to 250 k inhabitants	37	14.68
	city over 250 k inhabitants	93	36.9
Marital Status	single	49	19.44
	informal relationship	55	21.83
	married	139	55.16
	separated	9	3.57
Education	professional medical	3	1.19
	bachelor’s degree	38	15.08
	master’s degree	192	76.19
	specialist	19	7.54
Duration of Employment	up to 5 years	90	35.71
	6–10 years	52	20.63
	11–20 years	70	27.78
	21–30 years	31	12.3
	Over 31 years	9	3.57
Workplace—Social Center	no	243	96.43
	yes	9	3.57
Workplace—State Healthcare—Infirmary	no	230	91.27
	yes	22	8.73
Workplace—Private Resort	no	161	63.89
	yes	91	36.11
Workplace—Self-Practice	no	149	59.13
	yes	103	40.87
Workplace—State Healthcare—Hospital	no	177	70.24
	yes	75	29.76
Workplace—Other	no	210	83.33
	yes	42	16.67
Living with an Elderly Person	yes	166	65.87
	no	86	34.13
Personal Contacts with Elderly People	never	19	7.54
	only on occasion	25	9.92
	occasionally	181	71.83
	yes, few times a week	27	10.71
Professional Contacts with Elderly People	yes	218	86.51
	no	29	11.51
	I have no occasion	5	1.98
Keeping contacts with Elderly People from Outside the Family	yes	168	66.67
	no	39	15.48
	I have no occasion	45	17.86
Relevance of Contacts with Elderly People	not important	3	1.19
	indifferent	39	15.48
	important	149	59.13
	very Important	61	24.21

**Table 2 ijerph-19-03994-t002:** Results of Bayesian robust linear regression with KOAP score as the dependent variable.

	β	SE	LI	UI
Intercept	−0.23	0.31	−0.84	0.38
Gender	0	0.07	−0.14	0.14
Age	0.46	0.27	−0.06	0.97
Place of residence	0.12	0.12	−0.11	0.35
Marital status—Single	0.02	0.14	−0.24	0.3
Marital status—Informal relationship	−0.01	0.14	−0.29	0.26
Marital status—Married	0.08	0.11	−0.13	0.3
Education	0.38	0.29	−0.2	0.94
Duration of employment	−0.11	0.3	−0.7	0.49
Social center	0.01	0.17	−0.32	0.34
State health care—infirmary	−0.06	0.12	−0.3	0.17
Private resort	−0.08	0.07	−0.22	0.05
Self practice	0.01	0.07	−0.12	0.15
Hospital	0.06	0.08	−0.08	0.22
Other	0.08	0.09	−0.1	0.26
Living with an elderly person	0.03	0.07	−0.1	0.16
Personal contacts with elderly people	−0.09	0.21	−0.51	0.32
Professional contacts with elderly people (yes)	−0.13	0.16	−0.45	0.18
Professional contacts with elderly people (no)	−0.24	0.2	−0.6	0.13
Keeping contacts with elderly people from outside the family (yes)	−0.02	0.1	−0.21	0.17
Keeping contacts with elderly people from outside the family (no)	−0.08	0.12	−0.31	0.16
**Relevance of contacts with elderly people**	**1.24**	**0.22**	**0.82**	**1.66**
**JSE**	**0.12**	**0.06**	**0**	**0.25**
σ	0.91	0.04	0.83	1
R2	0.28	0.04	0.21	0.35

Note: β and SE, respectively, are the posterior mean and standard error of the mean. LI and UI are the lower and upper boundaries of the 95% credibility interval. The [n] symbol indicates the n-th coefficient of a sum-to-zero contrast for a categorical predictor. Bolded rows indicate statistically credible regression weights. σ and ν, respectively, are scale and normality parameters of the t distribution.

## Data Availability

The data presented in this study is available from the respective author upon request. The data is not publicly available due to the continuation of the project in this regard.

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
