# Peer review of "Empathy as a Factor Conditioning Attitudes towards the Elderly among Physiotherapists—Results from Poland"

_ijerph, 2022, doi:10.3390/ijerph19073994_

Round 1
Reviewer 1 Report
The research reports and analyzes the results of an internet study in which the respondents (n = 252) are self-selecting from a pool of physiotherapists in Poland. Mention should be made of the total number of physiotherapists in the population, the response rate, and the representativeness of the sample.
The authors are commended for a thorough methodologically driven study. The Bayesian approach is appropriate for the research
The aims of the study are presented in Lines 97- 104. However, the Discussion should more clearly articulate the results to the aims. Overall, were the aims achieved?
Seventeen variables (Table 1) are employed. It is not altogether clear the basis on which the independent variables were selected in the Introduction. Could the authors cite sources for the use of the variables?
I am surprised that the research shows that there was only one statistically credible relationship. Is this surprising given the number of independent variables.
The Discussion is missing a section on study limitations. Parts of the Discussion belong more properly in the Introduction as literature review (Lines 183 – 212).
Author Response
Dear Reviewer,thank you very much for your comments - they were very valuable to us. Our answers in the file and corrections in the manuscript.

Reviewer 2 Report
I have reviewed the manuscript and provided minor editing suggestions using sticky notes in the original submitted PDF document (attached herewith).

Author Response
Dear Reviewer, thank you very much for your comments - they were very valuable to us. Our answers in the file and corrections in the manuscript.

Reviewer 3 Report
- Please add the gap the study, inclusion criteria, and data analysis used in the abstract.
- Explain each factor relationship with attitude based on theoretical assumption and the gap of study.
- Please describe the sampling technique in the participants description.
- Describe validity and reliability of the instruments used in the study.
- Did researchers obtain permission for using the instruments?
- The margin of duration of employment should be different among categories. For example: How about respondents who had been working for five years? Were they categorized in the 1st or 2nd group?
- The authors should confirm whether KOAP is an instrument or a variable as mentioned in the line 166. The authors should write the
variable (not the instrument)
while displaying the results. JSE. - The main results are described in line 192-196. The authors should explain all results.
- Describe limitation of the study.
- Why do the authors mentioned about the previous studies results in the conclusion part?

Author Response

(The authors gave the same response as above.)

Reviewer 4 Report
This research is very interesting but there are quite a few major changes to be made.
Regarding the sample, the fact that more women than men participate may be an important factor that may influence the results due to cultural issues associated with the role of women. The sample that participated needs to be further defined, the field in which they practice the physiotherapy profession, if they work with the elderly, in public or private centers.
The discussion is very poor, it should be expanded.
It is necessary to expand the conclusions with more scientific evidence, compare with studies already carried out in other countries and if it has been evaluated with the same instrument (KOAP) or with different instruments.
It is advisable to expand the conclusions and include the limitations of the study.
Author Response

(The authors gave the same response as above.)

Round 2
Reviewer 1 Report
The authors make significant improvements to the manuscript.
I remain troubled that I cannot map a one-to-one correspondence from the literature to the variables used in the study.
As to sample size of 252, I would like to know how many invitations to the survey were issued and what the response rate was? Also, were more than 252 surveys returned? If so, how many were not used.
Author Response
Thank You for constructive comments in the review. Your comments have provided valuable information to improve its content and analysis.

Reviewer 4 Report
The conclusion must be improved
Author Response
Thank You for constructive comments in the review. Your comments have provided valuable information to improve its content and analysis.

This manuscript is a resubmission of an earlier submission. The following is a list of the peer review reports and author responses from that submission.
Round 1
Reviewer 1 Report
Ageism and empathy in healthcare professionals are two aspects of important social interest. In this matter, the study performed by the authors can result very attractive for the audience.
However, there are many aspects, some related to the structure and others to the design, which I strongly recommend to be improved prior to considering it as an article of IJERPH.
Introduction.-
In my opinion this part is well developed. It summarizes the social problem of ageism and its importance in the health sector. Authors stressed some aspects related to the pandemics. However, the importance of this issue precedes the pandemics. In my opinion, an aspect that is missing and should be covered, at least in part, is the concept of “active ageing”. Before the pandemics started in 2020, Europe was involved in a tendency to enhance this idea as the main solution for elderly people. Other aspects, such social life, were considered less important. I am wondering if a positive attitude towards “active ageing” can negatively influence the attitude that physiotherapists develop towards their patients (especially with patients who are not very physically active).
Hypothesis / Objectives.-
Authors present four aims that were not answered (and discussed) in their findings with the exception of empathy.
Methods.-
“Questionnaire” appears twice in one sentence (line 97). Authors can use another word, such as “scale”, “instrument” or “form”.
The description of the KAOP is incorrect (or confusing) in lines 103 – 104: “higher scores indicate a more positive attitude towards older people, and high scores indicate a more negative attitude towards older people”
Citing Hojat’s work, authors introduce “34” (Line 108). I assume that it is a mistake.
Procedure.-
This paragraph is quite simple and redundant in some parts. As inclusion criteria, authors report three characteristics: being adult (redundant), with no upper age limit (does it imply that there are retired professionals in the sample group? This is not clear because in line 88 it was explicitly mentioned that all participants were professionals in active). Additionally, in the socio-demographic form, authors collected sensitive information, such as: “professional contacts with elderly people” (29 people answered “no”). I am wondering if those professionals were included later in the analysis. Since there is some ambiguity in the hypothesis, at this point I am not able to know if authors intentionally are considering all professionals in their study (those who work with elderly people and those who do not) or the study was performed for measuring attitudes toward elderly patients among those professionals who work with them.
Some variables of the socio-demographic form are lack of an explanation. For readers is difficult to know the type of information that those variables collect.
Data analysis.-
No information is reported regarding the reliability of the instruments used in this study (i.e. Cronbach’s alpha coefficients) neither for the KAOP or the JSE. In line 130, the authors state that a variable is statistically credible when 95% CI of the distribution excludes “0”. However, in one of the two variables (JSE) the IL is “0”. Therefore, why do authors include this variable as a predictor?
Results.-
Table 1 is too long and less informative. Frequency of variables studied are not proportional in the following cases: marital status, education, duration of employment, social center, other, professional contact with elderly people, and relevance of contacts with elderly people, with categories with less than 10 respondents.
Reliability is missing for the KAOP and the JSE. Descriptive analysis of the JSE is missing either.
The model (Table 2) includes all variables but only one shows credible regression weights (relevance of contacts with elderly people), and another is not clear (JSE). Variables without credible regression weights should be removed from the model (following a backward / forward regression design). Perhaps, without other variables, values can vary and the role of the JSE in the model will be clearer. Do authors consider a ceiling effect in the case of the variable “relevance of contacts…”? According to Table 1, authors are comparing an intercept of 3 cases, with categories of 39, 149, and 61 cases, respectively. I am also wondering if those who answered “not important” or “indifferent” in this item are included in the group of those who do not have professional contact with elderly patients (29 cases) or those who have an occasionally contact with this type of patients (5 cases).
Discussion.-
If finally JSE has to be removed from the model, outcomes observed are not very informative: professionals with positive attitudes toward elderly patients are the ones for whom the contact with those patients are important or very important. Such conclusion could be quite obvious and there is not so much to discuss. I kindly recommend the authors to go back into the valuable data collected and re-consider their strategy of analysis.
There are many aspects discussed which, in my opinion, are not related with the findings reported in this study and should be removed: lines 183-195.
Conclusions.-
This part must be re-written.
Author Response
Dear Reviewer,
Thank You for constructive comments in the review. Your comments have provided valuable information to improve its content and analysis. In the document below, we try our best to solve the reported problems. Our responses are shown in blue after your comments.
M.Podhorecka

Reviewer 2 Report
Dear Authors,
Please see attached file with my comments and suggestions.
Best regards

Author Response

(The authors gave the same response as above.)

Round 2
Reviewer 2 Report
Dear Authors,
Please see attached file.
Best regards
